# Pulmonary Fat Embolism Following Liposuction and Fat Grafting: A Review of Published Cases

**DOI:** 10.3390/healthcare11101391

**Published:** 2023-05-11

**Authors:** Yu-Ming Kao, Kuo-Tai Chen, Kuo-Chang Lee, Chien-Chin Hsu, Yeh-Cheng Chien

**Affiliations:** 1Division of General Surgery, Department of Surgery, Chi-Mei Medical Center, Tainan 71004, Taiwan; mikegao2568549@yahoo.com.tw; 2Emergency Department, Chi-Mei Medical Center, Tainan 71004, Taiwan; 890502@mail.chimei.org.tw (K.-T.C.); ozisun.tw@yahoo.com.tw (C.-C.H.); 3Emergency Department, Chi-Mei Medical Center Chiali Branch, Tainan 71004, Taiwan; iamgord@hotmail.com; 4Department of Biotechnology, Southern Tainan University of Technology, Tainan 71005, Taiwan

**Keywords:** fat embolism, liposuction, fat grafting, cosmetic surgery, complication, outcome

## Abstract

Background: One of the most severe complications of liposuction and fat grafting is pulmonary fat embolism (PFE). However, most healthcare workers are not familiar with PFE. We performed a systematic review to describe the details of PFE. Methods: PubMed, EMBASE, and Google Scholar were searched up to October 2022. Further analysis focused on clinical, diagnostic, and outcome parameters. Results: A total of 38 patients from 20 countries were included. Chest computed tomography (CT) yielded 100% accuracy in the diagnosis of PFE. All of the deceased died within 5 days after surgery, and in 76% of patients, onset of symptoms occurred within 24 h after surgery. The proportions of patients who required mechanical ventilation, had a cardiac arrest event, or died among all patients and among those whose onset of symptoms occurred within 24 h after surgery were 75%, 38%, and 33% versus 79%, 56%, and 43%, respectively. Conclusions: The earlier the onset of symptoms was, the more severe the clinical course was. Once a patient presents with PFE-related symptoms, surgery should be halted, supportive care initiated, and chest CT used to diagnose PFE. According to our review results, if a patient with PFE survives the initial episode without permanent sequelae, a complete recovery can be anticipated.

## 1. Introduction

Plastic surgery was initially utilized for congenital and accidental defects of patients, its applications have expanded to include cosmetic purposes [1,2,3]. According to a recent report by the American Society of Plastic Surgeons, there has been a 132% increase in cosmetic procedures between 2000 and 2016 [4]. At the same time, the prevalence of obesity has significantly increased since 1980, especially among children and young adults in developing middle-income countries [5]. Due to concerns about body image and the influence of social media, cosmetic surgery for obesity is becoming increasingly popular [1,6]. The increased demand for cosmetic surgery raises concerns about adverse events related to surgery. Although generally considered safe, these surgeries carry a risk of severe complications during and after the surgery [7]. Among the various surgical procedures, liposuction and fat grafting are two commonly performed cosmetic surgical treatments [8,9]. Pulmonary fat embolism (PFE) is one of the most severe complications of liposuction and fat grafting [10]. During surgery, damage to adipose tissue and perforation of small blood vessels produce lipid fragments, which enter the venous system and cause lung injury.

Timely diagnosis and treatment of PFE resulting from liposuction and fat grafting is challenging for healthcare workers. First, the symptoms of PFE are similar to many surgery-related complications, such as adult respiratory distress syndrome, pulmonary embolism, pulmonary edema, aspiration pneumonia, drug allergy, and transfusion-related acute lung injury. It is difficult to differentiate PFE promptly from these complications [11]. Additionally, appropriate treatments for these diseases vary and may counteract each other. Second, because patients with PFE present in acute respiratory distress or in an unstable hemodynamic state, they are often taken to the emergency department, which is not the institution where the surgery took place. The scarcity of reliable surgical history for a distressed patient increases the diagnostic difficulty. Third, PFE occurs sporadically, and most healthcare workers are not familiar with its clinical features and management, which might delay optimal treatment for patients who typically require urgent care.

Efficient characterization and management of PFE are essential. To date, scattered case reporting has yielded insufficient insight into the full epidemiological, clinical, diagnostic, therapeutic, and prognostic spectrum. Therefore, we performed a systematic review, the most extensive to date, to analyze the clinical manifestations of PFE. We anticipate that the collected data can provide a complete picture of PFE, offer useful suggestions to healthcare workers, and improve the management of patients with PFE.

## 2. Methods

### 2.1. Data Sources and Search Strategy

The study did not involve human subjects or chart review; therefore, an institutional review board approval was not required. To identify related studies, the researchers conducted a search on PubMed, EMBASE, and Google Scholar from the earliest record up to October 2022, with no restrictions on language or country. Non-English-language manuscripts were translated using Google Translate (Mountain View, CA, USA). The researchers used various keywords combined with “fat embolism” for the search, including liposuction, lipectomy, lipoplasty, lipolysis, fat graft, mammaplasty, breast reconstruction, and gluteal augmentation. The reference lists of the included studies were also considered as additional sources.

### 2.2. Eligibility Criteria and Study Selection

Only case series and cases reports were discovered during the literature search. We reviewed the selected cases and included those that fulfilled at least one of the following criteria:(1)Pathological findings during autopsy or bronchoalveolar lavage revealed positive results for PFE [10].(2)Patients either presented with pulmonary symptoms or showed classical findings of PFE on images of chest computed tomography (CT) [12,13].

### 2.3. Data Extraction

Data collected from the reviewed studies included the published year, authors, countries, sex, age, comorbidities, types and body parts of surgery, presenting symptoms on hospital admission, laboratory tests, imaging studies, pathological reports, occurrence of cardiac arrest event, applied treatments, requirement for mechanical ventilation, complications, hospital course, requirement for intensive care, presence of extrapulmonary thromboembolism, sequelae of PFE, and mortality. 

Two authors (Y.M.K. and Y.C.C.) independently reviewed the titles and abstracts and selected relevant manuscripts for full-text review, resolving disagreements by consensus or by a third reviewer (K.T.C., one of the study authors).

The main variables of interest were the prevalence of complications, extrapulmonary thromboembolism, and permanent organ failure/disability, with complications defined as concomitant diseases resulting from surgery or hospitalization, extrapulmonary thromboembolisms defined as arterial or venous thromboembolisms during hospitalization involving an organ other than the lung, and permanent organ failure/disability defined as irreversible organ failure and long-term disability after hospital discharge.

## 3. Results

### 3.1. Characteristics of Enrolled Studies

We identified 184 studies from the database search and 207 studies from reference lists, resulting in a total of 391 studies. A total of 52 studies were excluded as duplicates. The titles and abstracts of the remaining 340 studies were reviewed. After excluding 251 studies, the authors meticulously reviewed the remaining 88 studies. Of these, 52 were excluded. Among these 52 excluded studies, 5 were meta-analyses, 8 involved allograft-related pulmonary embolism, 4 involved blood-clot-related pulmonary embolism, and 35 involved fat embolisms without pulmonary involvement. Finally, 36 studies (33 in English, 1 in Japanese, 1 in French, and 1 in Portuguese) involving 39 patients met our search criteria. We excluded the case of one patient in a collected study because it contained no evidence of PFE [14]. A total of 38 patients were included in our review. A flow diagram of the search and identification strategy is presented in Figure 1. Patient characteristics are presented in Table 1.

The study included a total of 38 patients, of which 84% were women and 82% had no comorbidities. The age of patients ranged from 21 to 82 years. Liposuction was performed in all studies, and fat grafting was performed in 34% of studies. It is impossible to distinguish the risk of PFE between patients with liposuction alone and those who also received fat grafting. In most cases, the dimensions of the cannula used during surgery were not recorded. The volume of solution aspirated during liposuction varied widely, ranging from 35 to 10,000 mL, and the most frequently treated body parts were the abdomen/flank, lower limbs, buttocks, and breast/chest (as shown in Table 2).

In total, 16 patients were in the United States, 4 were in Brazil, 2 were in France, 2 were in the United Kingdom, and 1 patient each was reported in Canada, China, Colombia, Egypt, Guatemala, India, Japan, Korea, Mexico, Romania, Saudi Arabia, South Africa, Sweden, Switzerland, Taiwan, and Vietnam. 

### 3.2. Symptoms, Laboratory Tests, and Diagnostic Measurements of Enrolled Patients

Common symptoms included dyspnea, hypotension, tachycardia, hypoxia, and altered mental state. A total of 21% of patients presented initially with cardiac arrest events. The time to symptom onset ranged from 0 to 13 days after surgery. Most patients (76%) had onset of symptoms within 24 h after surgery, and the time to symptom onset was not reported for one patient.

In terms of laboratory tests, all patients had decreased PaO_2_/FiO_2_ ratios (PaO_2_/FiO_2_ ≤ 200). Additionally, leukocytosis, anemia, thrombocytopenia, and elevation of D-dimer were frequently observed abnormalities. 

Autopsy, chest CT, and bronchoalveolar lavage yielded high accuracy in the diagnosis of PFE. Microscopic examination during autopsy typically shows multiple adipose tissue emboli in pulmonary arteries [40]. Chest CT scan commonly reveals diffuse mixed ground-glass and consolidative opacities involving both lungs. Bronchoalveolar lavage fluid typically contains blood-tinged secretions in the airways, and microscopic examination of the specimens typically reveals the presence of lipid-laden macrophages [13]. Before 1997, pulmonary angiograms revealed multiple irregular peripheral defects in the pulmonary arterial tree with subsegmental occlusions indicating micro emboli in three patients [16,19,20]. The majority of reported plain chest X-ray (CXR) findings revealed opacifications of bilateral lung fields, similar to the presentation of pulmonary edema or adult respiratory distress syndrome. However, 16% of the reported CXR findings were negative. In the acute stage, 67% of echocardiograms revealed abnormalities, including global hypokinesia, dilated right ventricle, or signs of pulmonary hypertension [28,34]. Symptoms, laboratory abnormalities, and various diagnostic measurements are listed in Table 3.

### 3.3. Treatment

The mainstay of treatment for patients with PFE is supportive care, which includes oxygen supplementation, ventilator support, and inotropic agents to manage hypoxia and hypotension. Antibiotics were used to prevent or treat wound infection and pneumonia. Four patients underwent corticosteroid therapy. In 2022, Wolfe reported a patient with severe PFE who underwent extracorporeal membrane oxygenation and survived [47]. Before the definite diagnosis of PFE, a few patients underwent intravenous heparin infusion to treat suspected pulmonary embolism based on their clinical picture and image findings. In an extreme case, a patient underwent combined intravenous streptokinase infusion, pulmonary artery catheter embolectomy, and lobectomy for suspected pulmonary embolism [16].

### 3.4. Resolution Time for Pulmonary Embolism

Hospitalization time ranged from 3 to 100 days [21,27]. Follow-up CXRs performed 2 days after hospitalization revealed resolution of abnormalities in two patients [17,21]. Follow-up chest CT performed 7, 10, and 10 days after hospitalization revealed resolution of abnormalities in four patients [21,24,34].

### 3.5. Outcome

The outcome of a patient was not described in two studies [31,44]. Therefore, 36 patients were included for outcome analysis. In total, 12 patients died, all within 5 days after surgery. Among the 37 patients, 76% required mechanical ventilation, 38% had cardiac arrest events, and 33% died. Among patients whose symptoms onset within 24 h after surgery, 75% required mechanical ventilation, 38% had cardiac arrest events, and 33% died (Figure 2).

Extrapulmonary thromboembolisms were commonly discovered during autopsy and imaging studies. Among the extrapulmonary thromboembolisms that were discovered, four were cerebral embolisms, two were retinal embolisms, two were lower limb venous thromboembolisms, one was a renal embolism, and one was a spleen embolism. However, among the 24 patients who survived, only 3 (13%) had extrapulmonary thromboembolisms. Complications, including three wound infections, one lung infection requiring lobectomy, one hypoxic encephalopathy, and one acute renal failure, were present in six patients (25%). Extrapulmonary thromboembolisms and complications were present in three patients (13%) with permanent organ failure and disability, including one case of blindness due to a retinal fat embolism, one disability related to cerebral infarction, and one renal failure requiring long-term hemodialysis. Figure 2 shows the incidences of mortality, cardiac arrest events, mechanical ventilation, extrapulmonary thromboembolism, complications, and permanent organ failure/disability of varied patient groups.

## 4. Discussion

### 4.1. Demographics and Diagnosis

The patients included in the studies were from 19 different countries located on different continents, indicating that liposuction and fat grafting surgeries are performed worldwide. The age range of patients also varied widely. The wide range of districts and ages demonstrated that PFE caused by liposuction and fat grafting is a threateningly global and general health issue. This issue also presents a challenge for healthcare workers to diagnose and offer appropriate care to patients with PFE. Liposuction and fat grafting were performed on various body parts, suggesting that PFE is not limited to specific body parts. Nearly one-third of patients died, and more than three-quarters of patients required artificial airway and mechanical ventilation, indicating that PFE is a devastating complication of liposuction and fat grafting. If a healthcare worker encounters a patient with unknown respiratory distress, circulatory instability, or neurological abnormalities during or after liposuction/fat grafting, PFE should be considered and investigated meticulously.

Fat embolism syndrome is a multisystem condition. Commonly used diagnostic criteria include those proposed by Gurd and by Schönfeld [31]. However, patients with PFE often experience rapid deterioration of their condition. The collection of adequate diagnostic criteria for a patient exhibiting signs of an emergency situation can be challenging, yet early and accurate recognition of PFE is essential for effective treatment. Treatment should be initiated immediately upon establishment of a diagnosis, and delays caused by unnecessary diagnostic tests should be minimized, while potential harmful therapy should be avoided. Accordingly, early application of imaging studies, especially chest CT, for patients exhibiting signs of PFE may be the best strategy for quickly obtaining an accurate diagnosis. Chest CT also enables the identification of other pulmonary and cardiovascular diseases with symptoms analogous to PFE.

Most patients with PFE undergo CXR examinations, and pulmonary abnormalities are identified in 90% of these scans. However, it can be challenging to distinguish between PFE-related abnormalities and those related to pulmonary edema, adult respiratory distress syndrome, or pneumonia, and further evidence is required to confirm the presence of PFE. Some patients are unable to receive chest CT scans due to an allergy to the contrast agent, poor renal function, or an unstable hemodynamic state. In these cases, echocardiography can be a useful alternative diagnostic tool. Signs of right heart overload and global hypokinesia are typical findings related to PFE, although these findings are not specific to PFE and can also be seen in patients with pulmonary thromboembolism. Patients with PFE and pulmonary thromboembolism both benefit from ventilation and circulation support, and healthcare workers can stabilize patients and arrange further diagnostic measurements to distinguish between the two conditions. 

Patients who are stabilized with an artificial airway and mechanical ventilation may undergo bronchoalveolar lavage, which can provide sensitive and specific results for diagnosing PFE. The findings of bronchoalveolar lavage can exclude several other diseases that present similar symptoms to PFE and avoid unnecessary treatments on critically ill patients.

Most patients with PFE present with various and non-specific symptoms related to the lungs and heart. If these symptoms worsen rapidly or are accompanied by involvement of other systems, such as altered mental state, neurological deficit, or skin rash, PFE should be immediately considered and confirmed with the aforementioned diagnostic tests.

Although arterial blood gas analysis is both sensitive and specific in detecting PFE, a finding of PaO_2_/FiO_2_ ≤ 200 mmHg can be present in a variety of diseases. Arterial blood gas analysis can be used to identify severe clinical conditions that require meticulous care. Other laboratory tests may reveal abnormalities that are non-specific to PFE, which can be included as part of the diagnostic criteria to confirm the diagnosis of PFE [48].

### 4.2. Management of Pulmonary Fat Embolism and Complications

The review by Bayter-Marin et al. proposed an explanation for the various clinical manifestations of fat embolism based on whether the embolism was macroscopic or microscopic [49]. In cases involving macroscopic fat embolism, the presence of fat in the cardiac cavities leads to mechanical heart failure and subsequent occlusion of the pulmonary arteries. In cases involving microscopic fat embolism, small amounts of fat enter the bloodstream, affecting the pulmonary capillaries and other organs. Patients in many of the included studies experienced dyspnea or hypotension during surgery, followed by rapid deterioration of their condition. Therefore, to prevent more fat from entering the bloodstream, the surgical procedure should be halted immediately.

The mainstay of treatment for patients with PFE is supportive management, especially respiratory and hemodynamic support. Supportive management involves ensuring adequate oxygenation with oxygen supplementation and mechanical ventilation, if necessary, and achieving adequate circulation with fluid infusion, inotropic agents, and even mechanical circulatory support. Extracorporeal membrane oxygenation is a potential treatment approach for patients with severe PFE, providing both oxygenation and hemodynamic support.

The treatment of patients with PFE may be complicated by infection and extrapulmonary thromboembolism. Surgical wound infection and ventilator-associated pneumonia are common infectious complications. Therefore, a large number of patients receive antibiotics. Extrapulmonary thromboembolism may result from fat embolism itself or develop later due to immobilization-related blood clot formation. Magnetic resonance imaging of the brain and CT of the torso are indicated in patients presenting with neurological deficits or organ damage. Thromboembolism prophylaxis should be initiated if indicated for immobilized patients [50].

### 4.3. Hospital Course and Outcome

Most patients with PFE have a rapid and severe clinical course. About 69% of patients experience symptoms during or within 24 h after surgery, and more than 90% of deaths occur within 5 days after surgery. Patients who develop symptoms within 24 h after surgery are more likely to require mechanical ventilation, have more cardiac arrest events, and are more likely to die than patients whose symptoms appear 24 h after surgery or later. This finding suggests that the earlier the onset of symptoms, the more severe the clinical course. 

However, if a patient with PFE survives the initial episode and resuscitation does not result in permanent sequelae, such as hypoxic encephalopathy or renal failure, the symptoms of PFE usually resolve spontaneously, and a complete recovery can be expected. Among the patients reviewed, abnormalities identified by CXR resolved within a day in one case, and lesions on chest CT resolved 7 to 10 days after hospitalization. Due to the lack of sufficient data, we cannot be certain whether the resolution of abnormalities on images corresponds exactly to the improvement of the patient’s symptoms. Nevertheless, follow-up pulmonary images demonstrate the potential for therapeutic decisions for patients with PFE.

### 4.4. Strengths and Limitations

This literature review documents cases of PFE from 19 countries worldwide and is the most comprehensive to date. The included studies are globally representative, and the conclusions drawn are useful for healthcare workers worldwide. The study provides a detailed review and analysis of the clinical manifestations of PFE, including details of surgery, onset timing and symptoms, laboratory findings, abnormalities on images and examinations, patient management strategies, hospital courses, and outcomes. Healthcare workers can derive a comprehensive understanding of PFE from this review.

However, this manuscript retains the unavoidable limitations of a review article. First, most reported cases were severe, and information about patients with mild symptoms was omitted. Since symptoms of PFE might resolve with supportive management, patients with mild symptoms might not exhibit clinical significance. Healthcare workers must identify and manage severe PFE early and provide proper treatment. Second, bias was present in the case reports, and data from these studies do not provide as convincing evidence as that of a clinical trial. However, severe PFE is not common in clinical practice, making it challenging to obtain data with enough power in a well-designed trial. Reviewing reported cases may be the most practical approach for presenting the details of PFE. Third, the quality of care for patients with PFE in different reported cases did not follow a universal rule. Therefore, the heterogeneity of hospital courses and outcomes of patients may be derived from different care instead of the influences from different treatments.

## 5. Conclusions

PFE can occur in any body part undergoing surgery involving liposuction or fat grafting. The earlier the onset of symptoms, the more severe the clinical course tends to be. If a patient undergoing such surgery presents with PFE-related symptoms, such as dyspnea, hypotension, hypoxia, or altered mental state, the first step is to stop the surgical procedure to prevent more fat from entering the blood vessels. The next step is to initiate supportive care to ensure acceptable oxygenation and adequate circulation. The diagnostic criteria for a patient exhibiting signs of an emergency situation may not be appropriate for rapid diagnosis of PFE. Therefore, the liberal use of chest CT to prove or exclude the presence of PFE is warranted. According to our review results, if a patient with PFE survives the initial episode without permanent sequelae, the symptoms usually resolve spontaneously, and a complete recovery can be anticipated.

## Figures and Tables

**Figure 1 healthcare-11-01391-f001:**
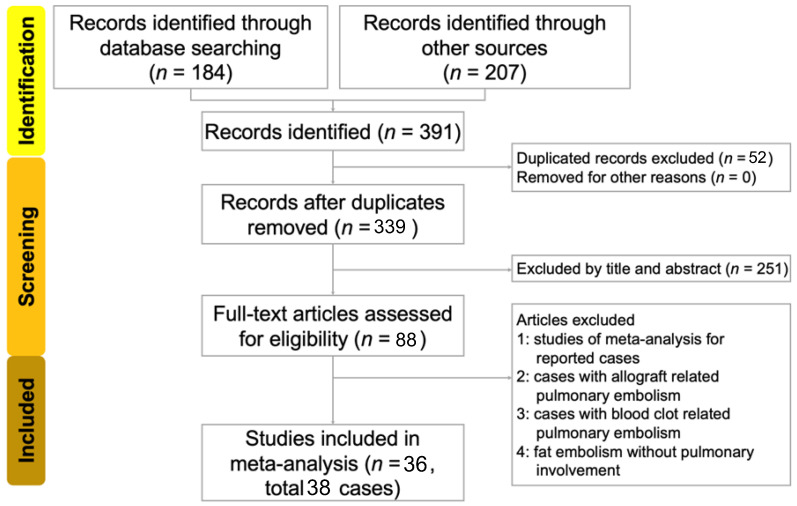
Flow diagram for the search and identification of included studies and patients.

**Figure 2 healthcare-11-01391-f002:**
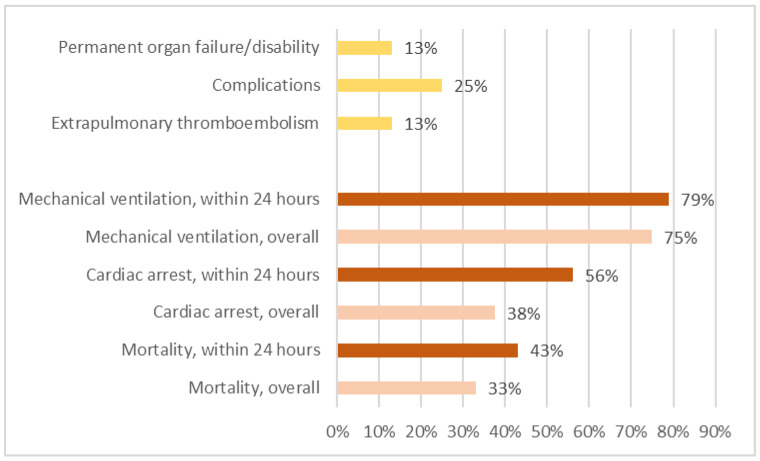
Mortality, cardiac arrest events, and mechanical ventilation rates for all patients and for patients with symptoms onset within 24 h after surgery. We excluded the deceased and calculated rates for extrapulmonary thromboembolism, complications, and permanent organ failure/disability.

**Table 1 healthcare-11-01391-t001:** Reviewed studies and patients’ characteristics.

Year, Author	Country	Sex	Age	Surgery	Cardiac Arrest	Mechanical Ventilation	Mortality
1983, Hunter GR [15]	US	F	37	Liposuction	No	Unknown	Alive
1986, Christman KD [16]	US	F	56	Liposuction	Yes	Yes	Dead
1988, Ross RM [17]	US	F	44	Liposuction	No	Yes	Alive
1990, Boezaart AP [18]	South Africa	F	39	Liposuction	No	Yes	Alive
1990, Laub Jr DR [19]	US	F	51	Liposuction	No	No	Alive
1997, Currie I [20]	Canada	F	69	Liposuction, fat grafting	Yes	Yes	Dead
1998, Fourme T [21]	France	F	29	Liposuction	No	No	Alive
1999, Folador JC [22]	Brazil	F	40	Liposuction	No	No	Alive
1999, Scroggins C [23]	US	F	54	Liposuction	No	Yes	Alive
2002, Platt MS-1 [14]	US	F	82	Liposuction	Yes	Yes	Dead
2002, Platt MS-2 [14]	US	M	50	Liposuction	Yes	Yes	Dead
2006, Rothmann C [24]	France	F	24	Liposuction	No	Yes	Alive
2007, Wessman DE [25]	US	M	31	Liposuction	No	No	Alive
2008, Costa AN [26]	Brazil	M	53	Liposuction, fat grafting	No	Yes	Alive
2011, Erba P [27]	Switzerland	F	46	Liposuction	No	Yes	Alive
2011, Gleeson CM [28]	UK	F	37	Liposuction, fat grafting	Yes	Yes	Dead
2012, Shiffman MA [29]	US	F	40	Liposuction, fat grafting	Yes	Yes	Dead
2013, Zeidman M [30]	US	F	24	Liposuction	No	Yes	Alive
2014, Cohen L [31]	US	F	58	Liposuction	Unknown	Unknown	Unknown
2014, Hostiuc S [32]	Romania	F	56	Liposuction	Yes	Yes	Dead
2015, Astarita DC [33]	US	F	42	Liposuction, fat grafting	Yes	Yes	Dead
2015, Byeon SW [34]	Korea	M	21	Liposuction	No	Yes	Alive
2015, Cárdenas-Camarena L [35]	Colombia	F	37	Liposuction, fat grafting	Yes	Yes	Dead
2015, Fu X [36]	China	F	30	Liposuction, fat grafting	No	No	Alive
2015, Vidua RK [37]	India	F	39	Liposuction	Yes	No	Dead
2016, Souza RL [12]	Brazil	F	42	Liposuction, fat grafting	No	Yes	Dead
2017, Ali A [38]	UK	F	45	Liposuction	No	Yes	Alive
2017, Sasaki Y [39]	Japan	F	29	Liposuction	No	Yes	Alive
2017, Zilg B [40]	Sweden	M	31	Liposuction, fat grafting	Yes	Yes	Dead
2019, Peña W [41]	Mexico	F	41	Liposuction, fat grafting	Yes	Yes	Alive
2019, Saon MD [42]	US	F	52	Liposuction	No	No	Alive
2020, Recinos S [43]	Guatemala	M	37	Liposuction, fat grafting	Yes	Yes	Alive
2021, Kadar A [13]	US	F	26	Liposuction	No	Yes	Alive
2022, Fonseca EKUN [44]	Brazil	F	32	Liposuction	Unknown	Unknown	Unknown
2022, Foula AS [45]	Egypt	F	29	Liposuction	Yes	Yes	Alive
2022, Pham MQ [46]	Vietnam	F	37	Liposuction	No	No	Alive
2022, Wolfe EM-1 [47]	US	F	28	Liposuction, fat grafting	No	Yes	Alive
2022, Wolfe EM-2 [47]	US	F	26	Liposuction, fat grafting	No	Yes	Alive

**Table 2 healthcare-11-01391-t002:** Demographic data, types of surgery, and body parts.

Demographic Data	Percentage	Body Parts	Percentage
Age (years)	39.0 (30.3–49.0) *	Abdomen/flank	21 (55%)
Sex (female)	32 (84%)	Lower limbs	14 (37%)
Comorbidity	7 (18%)	Buttocks	12 (32%)
Surgery		Breast/chest	9 (24%)
Liposuction	38 (100%)	Upper limbs	3 (8%)
Fat grafting	13 (34%)	Head/neck	2 (5%)
Others #	3 (8%)	Penis	2 (5%)

* Median (interquartile range). # Others: repairs of diastasis rectus in two patients, and hysterectomy and oophorectomy in one patient.

**Table 3 healthcare-11-01391-t003:** Symptoms, laboratory tests, and diagnostic measurements of the included patients. In some studies, patient characteristics were not described; therefore, the sum for each item may not be 40.

Symptoms	Percentage	Diagnostic Measurements	Percentage
Dyspnea	21 (55%)	Examinations	
Hypotension	16 (42%)	CT scan	18/18 (100%)
Tachycardia	14 (37%)	CXR	16/19 (84%)
Hypoxia	12 (32%)	Echocardiogram	8/12 (67%)
Altered mental state	9 (24%)	Bronchoalveolar lavage	4/4 (100%)
Cardiac arrest	8 (21%)	Autopsy	10/10 (100%)
Fever	7 (18%)	Pulmonary angiogram	3/3 (100%)
Skin rash/petechiae	5 (13%)	Laboratory tests	
Chest pain	4 (11%)	PaO_2_/FiO_2_ ≤ 200 mmHg	18/18 (100%)
Cyanosis	4 (11%)	White cell count > 1000, < 4000/μL	9/11 (82%)
Cough	3 (8%)	Hemoglobin < 12 g/dL	8/14 (57%)
Hemoptysis	3 (8%)	Platelet < 150,000/μL	5/11 (45%)
Syncope	2 (5%)	Creatine > 1.2 g/L	2/5 (40%)
Bradycardia	2 (5%)	T bilirubin > 1.2 g/L	2/3 (67%)
Neurologic deficit	2 (5%)	D-dimer > 500 mg/L	4/4 (100%)

## Data Availability

The source codes and data presented in this study are available on reasonable request to the corresponding author (K.T.C.). The raw data are not publicly available because of a data protection policy for patient data and/or patent.

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
