# Peer review of "Pulmonary Fat Embolism Following Liposuction and Fat Grafting: A Review of Published Cases"

_healthcare, 2023, doi:10.3390/healthcare11101391_

Round 1

Reviewer 1 Report

Manuscript ID: 2296588   Dear Authors,   I greet the authors of this manuscript for selecting this interesting work. Grafting has gained popularity in recent years recently, people should get insights on the adverse effects and the treatment approaches regarding this.   My comments are as follows:   1. In several places sentences are incomplete, please do correct them.   2. Carry out language correction for the entire manuscript.
3. Check the correct usage of the term "bloody secretion"
4. Tables 2 and 3 :  Add column names.   5. Writing the methods section with sub-headings (Search strategy, Eligibility criteria, Study selection and Data extraction) will enhance the lucidness.
6. In some places, there were no coherence in the writing, try to organize your data without unnecessary barriers.

7. This section was a bit ambiguous "Treatment, resolution time for pulmonary embolism, and outcome" refine and rewrite those sentences.

8. Correct the formatting mistakes.   9. Clearly write about the supportive management, treatment options as well.   10. Examine the sentence "If a patient with PFE survives the initial episode without permanent sequelae, the symptoms of PFE usually resolve spontaneously and a complete recovery can be anticipated" before revising this manuscript.

Author Response

Please find it in the attachment.

Reviewer 2 Report

The authors described a very dangerous condition affecting patients after liposuction and lipofilling. Few works are available in literature, hence the pulmonary fat embolism definitively deserves more attention in medical literature. The manuscript discuss a very interesting topic.

After revising the manuscript we would like to ask the authors few questions.

1) even considering that this is a review of case series and case reports, which do yoou think is more at risk of PFE between lipofilling+liposuction or liposuction alone?

2) the cannulas' dimensions are not described in the review; did you get more information from the mentioned works regarding the correlation between the diameter of the cannulas used and the risk of PFE?

3) did you investigated the fluctiation of DDimer among the labs tests? althoug it is a non-specific albeit sensitive parameter, do you think it would be appropriate to consider it in PFE work-up?

4) which is the therapeutic role of LMWH or analogues in PFE? this information is not considered

Icona di Verificata con community

Author Response

Please find it in the attachment.

Reviewer 3 Report

The authors perform a literature review on Pulmonary Fat Embolism and identify 45 patients.  A few suggestions for improvement.

1)  I believe due to the retrospective nature of this paper, the authors should be very careful about the statements they make.  For example, they state that CT scan is 100% diagnostic.  Obviously, this is a highly selected series and we don't actually know the baseline rate as not all patients having these procedures get CT scans.

2)  The authors should shorten the introduction and Disucssion and focus on what this study adds to the already extensive literature.

3)  There are excess significant digits throughout the manuscript for only 41 patients.  For example 83.6 would better be reported as 84.

4)  A careful edit by a native English speaker would be very helpful.  

Author Response

Please find it in the attachment.

Round 2

Reviewer 1 Report

Dear Authors,

You have revised this manuscript, incorporating my comments.

I have a of couple suggestions:

1. Start titles with capital letter for ex: "percentage" in table 3

Regarding the final comment of mine from the previous review:
 "If a patient with PFE survives the initial episode without
permanent sequelae, the symptoms usually resolve spontaneously, and a complete recovery can be anticipated".
 I wanted you to recheck those sentences since it was a statement by you about the complete recovery of a patient from PFE.

In this revision:
You said that you had examined that carefully. Even so, in my opinion it is better to add a sentence like "according to our review results" before the start of this sentence "If a patient with PFE survives the initial episode without permanent sequelae......... a complete recovery can be anticipated". Therefore, it will become a proper way of concluding your study. (Since it was a result of a study up to October 2022 and You have included other study limitations too.)

But, It's only a suggestion, you can go with the same old sentences if you are confident about what you are presenting.

Author Response

Reviewer 1

Dear Authors,

You have revised this manuscript, incorporating my comments.

I have a of couple suggestions:

  1. Start titles with capital letter for ex: "percentage" in table 3

Response: Thank you for your suggestion. We added capital letter in table 2 and 3.

Regarding the final comment of mine from the previous review:

 "If a patient with PFE survives the initial episode without

permanent sequelae, the symptoms usually resolve spontaneously, and a complete recovery can be anticipated".

 I wanted you to recheck those sentences since it was a statement by you about the complete recovery of a patient from PFE.

In this revision:

You said that you had examined that carefully. Even so, in my opinion it is better to add a sentence like "according to our review results" before the start of this sentence "If a patient with PFE survives the initial episode without permanent sequelae......... a complete recovery can be anticipated". Therefore, it will become a proper way of concluding your study. (Since it was a result of a study up to October 2022 and You have included other study limitations too.)

But, It's only a suggestion, you can go with the same old sentences if you are confident about what you are presenting.

Response: Thank you for the remarkable comment.

We added “According to our review results, “ before the sentence “if a patients with PFE survives the initial episode without permanent sequelae…..”. in the Abstract section and in the Conclusion section.

Reviewer 3 Report

Thanks for your revisions.

Author Response

Reviewer 3

Thanks for your revisions.

Response: Thank you for your suggestions. We learned a lot from your comments.